

# Relative evolutionary rate inference in HyPhy with LEISR

Stephanie J. Spielman and Sergei L. Kosakovsky Pond

Institute for Genomics and Evolutionary Medicine, Temple University, Philadelphia, PA,
United States of America

## ABSTRACT

We introduce LEISR (Likehood Estimation of Individual Site Rates, pronounced "laser"), a tool to infer relative evolutionary rates from protein and nucleotide data, implemented in HyPhy. LEISR is based on the popular Rate4Site (*Pupko et al., 2002*) approach for inferring relative site-wise evolutionary rates, primarily from protein data. We extend the original method for more general use in several key ways: (i) we increase the support for nucleotide data with additional models, (ii) we allow for datasets of arbitrary size, (iii) we support analysis of site-partitioned datasets to correct for the presence of recombination breakpoints, (iv) we produce rate estimates at all sites rather than at just a subset of sites, and (v) we implemented LEISR as MPI-enabled to support rapid, high-throughput analysis. LEISR is available in HyPhy starting with version 2.3.8, and it is accessible as an option in the HyPhy analysis menu ("Relative evolutionary rate inference"), which calls the HyPhy batchfile LEISR.bf.

## INTRODUCTION

Evolutionary rate inference is a fundamental analysis in computational molecular evolution (*Echave, Spielman & Wilke, 2016*). A widely-used tool for inferring evolutionary rates from phylogenetic protein data is Rate4Site, which exists both as a server and a command-line tool (*Pupko et al., 2002*). Although this method has proven extremely useful over the years, garnering nearly 500 citations, Rate4Site has several limitations: (i) it cannot analyze more than ~200–300 sequences because of numerical underflow issues; (ii) it often fails to converge to stable estimates if data are sufficiently complex even with relatively few (25–100) sequences; (iii) it accepts primarily protein data only and has limited nucleotide utility; and (iv) it only infers rates for sites which are not gaps in either the first sequence seen in the input file or a specified reference sequence and ignores remaining sites. As the number of available genomic sequences continues to rapidly expand, tools to analyze large data sets of any genomic type (protein and nucleotide), are needed.

To this end, we introduce a generalization of Rate4Site, which we term "LEISR" (**L**ikehood **E**stimation of **I**ndividual **S**ite **R**ates). LEISR is available as part of a

Corresponding author
Stephanie J. Spielman,
stephanie.spielman@temple.edu

**Table 1** **Nucleotide and Protein models, both generalist and specialist, available for use in LEISR, as of HyPhy version 2.3.8.** Future HyPhy versions are expected to include more models. Users can alternatively define and fit other parametric and empirical models with the use of HBL, the HyPhy batch language.

| Data type | Models |
|---|---|
| Nucleotide | GTR (*Tavare, 1984*), HKY85 (*Hasegawa, Kishino & Yano, 1985*), JC69 (*Jukes & Cantor, 1969*) |
| Protein, Generalist | LG (*Le & Gascuel, 2008*), WAG (*Whelan & Goldman, 2001*), JTT (*Jones, Taylor & Thornton, 1992*), JC69 (*Jukes & Cantor, 1969*) |
| Protein, Specialist | mtMet (*Le, Dang & Le, 2017*), mtVer (*Le, Dang & Le, 2017*), gcpREV (*Cox & Foster, 2013*), HIV B/W (*Nickle et al., 2007*) |

[1]Earlier versions of HyPhy (specifically, $\geq$ 2.3.6) also contain the LEISR method, although those pre-release versions will have reduced functionality relative to the LEISR implemented in HyPhy version 2.3.8.

[2]We note that HyPhy contains several robust methods (including FEL (*Kosakovsky Pond & Frost, 2005*), SLAC (*Kosakovsky Pond & Frost, 2005*), and FUBAR (*Murrell et al., 2013*)) for inferring site-wise evolutionary rates from codon data; see *Spielman, Wan & Wilke (2016)* for recommendations specifically on codon-level rate inference

leading molecular evolution inference platform HyPhy starting with version $\geq$ 2.3.8.[1] LEISR can be used to infer relative evolutionary rates from either nucleotide or protein data, thereby providing a flexible and fast platform for rate inference that may complement codon-level rate inference.[2] LEISR has been successfully tested with alignments containing up to 10,000 sequences, several orders of magnitude beyond what Rate4Site can fit. In addition, LEISR is MPI-enabled to support rapid inference from datasets with many sites by distributing optimization tasks to multiple compute nodes. Like other methods in HyPhy, LEISR allows users to provide partitioned alignments, with separate phylogenies for each partition, to correct for the effect of recombination during rate inference. Such partitioned alignments can be obtained, for example, with the method GARD (*Kosakovsky Pond et al., 2006*) in HyPhy.

## APPROACH

As input, LEISR requires a phylogeny and multiple sequence alignment, and its algorithm proceeds in two steps. It first obtains estimates of alignment-wide branch lengths (considering the input topology as fixed) under a user-specified substitution model (Table 1), and infers at each site a scaling parameter, $r_s$, that is used to uniformly scale all the branch lengths of the partition-specific tree at the site. $r_s$ can therefore be interpreted as the evolutionary rate at a specific site relative to the alignment-wide mean rate.

Rate4Site offers two statistical frameworks for rate inference: maximum-likelihood (ML) (*Pupko et al., 2002*) and empirical Bayes (*Mayrose et al., 2004*). Their ML framework is a "fixed effects" approach where a separate rate parameter is inferred at each site. Their empirical Bayes framework, by contrast, employs a "random effects" approach where rates are drawn from a prior gamma distribution. The LEISR implementation is analogous to the Rate4Site ML approach. During LEISR's branch length optimization stage, users can specify whether to model rate variation, with, if chosen, either a discrete gamma distribution (*Yang, 1993*) or the general discrete distribution (GDD) (*Kosakovsky Pond & Muse, 2005*). Although we provide the option to consider rate variation, we encourage users to opt for no rate variation. Indeed, the desired behavior for this method is for *only* the relative site-wise rates to contain information about site-wise evolutionary rate heterogeneity. If branch length optimization considers rate variation, then this information will be "conflated" between these two parameters (branch lengths and site rates). In other

words, one can view Rate4Site and LEISR as non-parametric rate estimation methods, whereas gamma and GDD are parametric estimation methods, and layering the two would be inefficient.

As LEISR inference proceeds, HyPhy will write markdown-formatted (*MacFarlane, 2017*) status-indicators to the console, including the inferred site-wise maximum-likelihood rate estimates with the approximate 95% confidence interval (CI) obtained via profile likelihood. All final output is written to a JSON-formatted (*Crockford, 2006*) file, named as the input data file with the suffix .LEISR.JSON. Site-wise rates are stored in the top-level JSON field MLE, whose CONTENT field contains a row for each site's inferred rates. Individual values in each row correspond to information given in the HEADERS key. A general description of HyPhy output JSON contents is available from http://www.hyphy.org/ in the "Resources" tab.

Users are free to transform these rates in a manner that suits their given analyses. For example, Rate4Site computes a standard score for each site, and other applications have called for normalizing each rate by the mean (or median) gene-wide rate (*Jack et al., 2016*; *Sydykova et al., 2017*). This latter scheme re-scales the average gene rate as 1, lending a more intuitive interpretation to each site's rate, i.e., a rate of 2 indicates that a site evolves twice as quickly as does an average site, and a rate of 0.5 indicates that a site evolves half as quickly as does an average site. That said, in certain circumstances, empirical rate distributions may be overdispersed and zero-inflated. In such cases, we suggest to normalize by the median rather than the mean, should normalization be desired. We note that even raw rate estimates generated by LEISR are already defined relative to the jointly-inferred partition mean rate.

## RESULTS

We confirmed that LEISR yields comparable inferences to Rate4Site using simulations. For each of three random phylogenies with 25, 50, and 100 taxa each, we simulated 10 replicate alignments, each with 100 sites, under the WAG model of protein evolution (*Whelan & Goldman, 2001*). Tree lengths (sum of branch lengths) for each tree, respectively, were 13.85, 27.32, and 52.83, and all trees had a mean branch length of ∼0.27. Our simulations modeled rate heterogeneity among sites with a discrete gamma distribution with 20 categories and a shape parameter of 0.4. Each replicate number used the same model parameterizations for all three trees, i.e., replicate 1 employed the same model for 25, 50, and 100 taxa. Simulations were conducted using the Python simulation library pyvolve (*Spielman & Wilke, 2015*).

We then inferred relative evolutionary rates in LEISR in two modes: turning off rate heterogeneity during branch length optimization ("LEISR"), and specifying a four-category discrete gamma distribution during branch length optimization ("LEISR+G"). We again inferred rates in two modes in Rate4Site (specifically their ML algorithm): without rate heterogeneity during branch length optimization ("R4S") and with a four-category discrete gamma distribution during branch length optimization ("R4S+G"). While the default number of rate categories for this step in Rate4Site is 16, but Rate4Site failed with errors for all 100-taxa simulations. We therefore used four rate categories, to both achieve a fair comparison with LEISR and to ensure

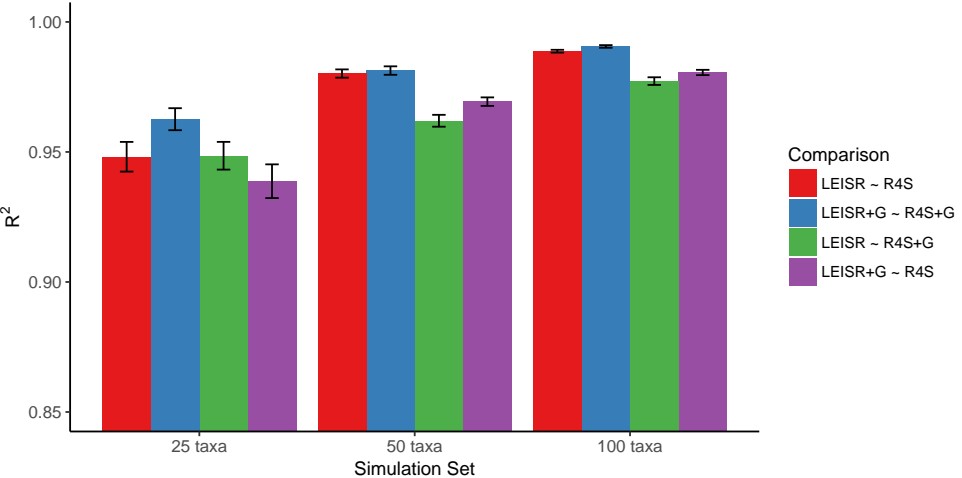

**Figure 1** Mean $R^2$ values (across 10 replicates) between inferred evolutionary rates across platforms and simulations. Bars represent the standard error of the mean. Note that the $y$-axis of this figure begins at $R^2 = 0.85$. All code to generate simulations and reproduce figures is available from https://github.com/sjspielman/leisr_validation.

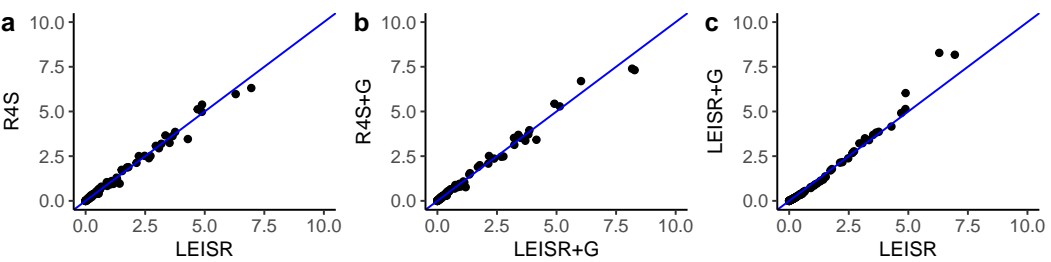

**Figure 2** Inferred evolutionary rates for a single simulation replicate with 100 taxa. The line shown in (A–C) is $y = x$. All code to generate simulations and reproduce figures is available from https://github.com/sjspielman/leisr_validation.

that Rate4Site could complete inferences. For those runs which completed, we observed comparable run times between LEISR and Rate4Site. Finally, for each alignment inference, we normalized rate estimates by dividing all rates by the mean site rate estimate, as described earlier.

In Fig. 1, we show $R^2$ values for Pearson's linear correlation between LEISR and Rate4Site inferences, computed across all simulations. The $R^2$ values further increase as the number of taxa increases, although even with 25 taxa the agreement is remarkably high. This trend is expected because the precision of inference for individual site rates will increase for larger samples with more taxa *Scheffler, Murrell & Kosakovsky Pond (2014)*. Figure 2 shows, for a single representative simulation replicate of 100 taxa, the relationship between inferred site rates across different methods and/or parameterization. Overall, these results demonstrate a nearly complete agreement between LEISR and Rate4Site, with rate inferences showing the closest agreement when the same option for branch length optimization was specified

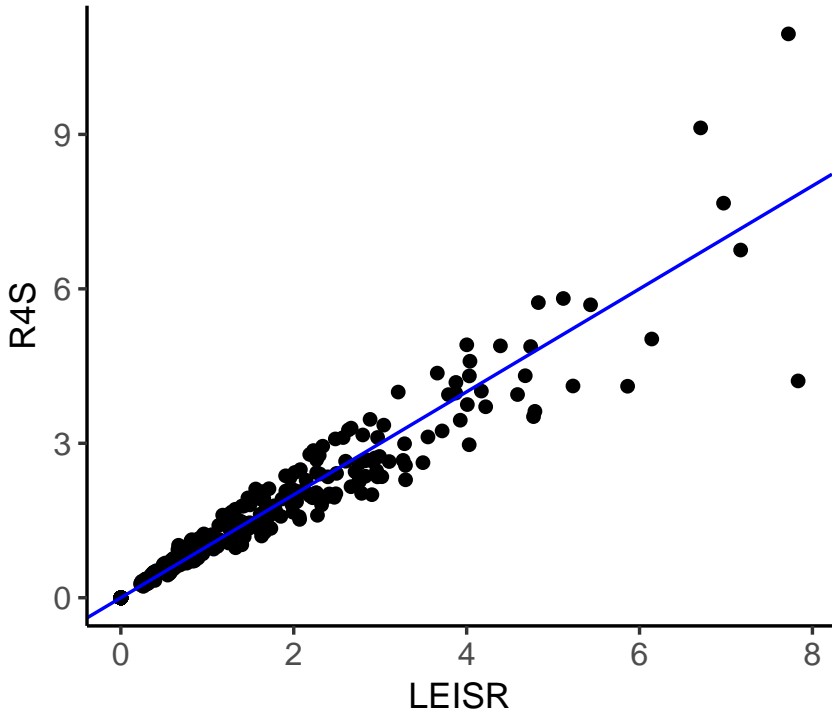

**Figure 3** **Inferred evolutionary rates with LEISR and Rate4Site on an empirical alignment of mammalian HRH1 receptors.** The line shown is $y = x$. Code to infer rates and reproduce this figure is available from https://github.com/sjspielman/leisr_validation.

(i.e., turned off or with a discrete gamma distribution). Although our simulations consisted relatively short gene sequences of only 100 sites, LEISR's use of a fixed effect approach means it will show similar accuracy for longer gene sequences. In addition, we found that nucleotide rate inferences under the JC69 model (the only currently available nucleotide model in the Rate4Site command line version) show similarly strong agreement between LEISR and Rate4Site.

Finally, we examined whether the comparability in rate estimates from simulated data extends to empirical data. We inferred rates using both LEISR and Rate4Site on an established mammalian protein alignment of HRH1 receptor (histamine receptor type 1) genes consisting of 23 sequences and 507 sites, where the phylogeny had a total tree length of 3.30 and a mean branch length of 0.077 (*Spielman & Wilke, 2013*; *Sydykova et al., 2017*). We specified the WAG model of evolution and branch length optimization without rate variation. Because Rate4Site only infers rates at sites which are not gaps in a reference sequence, we removed all sites which were gaps in the first sequence present in the the multiple sequence alignment before inference. This step resulted in a final alignment of 478 sites and ensured that rates from each platform were directly comparable.

As shown in Fig. 3, rate inferences between LEISR and Rate4Site on empirical protein data are extremely similar, with an $R^2 = 0.93$. This strong of agreement with 23 sequences is consistent with the observed $R^2$ values from the simulated datasets with 25 taxa

(Fig. 1), in spite of the overall fewer substitutions present in the empirical data related to the simulated data. We therefore find, using both simulated and empirical data, that LEISR provides a robust and reliable platform that can be used in the place of Rate4Site when dataset size and/or complexity preclude Rate4Site use, or when recombination is suspected.

## ACKNOWLEDGEMENTS

We encourage users who use LEISR to additionally cite Rate4Site (*Pupko et al., 2002*), which provides much of the intellectual basis and historical precedent for our implementation.

### Funding
This work was supported in part by grants R01 GM093939 (NIH/NIGMS), R01 AI134384 (NIH/NIAID), and U01 GM110749 (NIH/NIGMS). There was no additional external funding received for this study. The funders had no role in study design, data collection and analysis, decision to publish, or preparation of the manuscript.

### Grant Disclosures
The following grant information was disclosed by the authors:
NIH/NIGMS: R01 GM093939, U01 GM110749.
NIH/NIAID: R01 AI134384.

### Competing Interests
The authors declare there are no competing interests.

### Author Contributions
- Stephanie J. Spielman conceived and designed the experiments, performed the experiments, analyzed the data, contributed reagents/materials/analysis tools, wrote the paper, prepared figures and/or tables, reviewed drafts of the paper.
- Sergei L. Kosakovsky Pond contributed reagents/materials/analysis tools, wrote the paper, reviewed drafts of the paper.

### Data Availability
GitHub: https://github.com/sjspielman/leisr_validation.

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
