# Peer review of "Relative evolutionary rate inference in HyPhy with LEISR"

_PeerJ, doi:10.7717/peerj.4339_

## Round 0.1 · original submission · Minor Revisions

Nice manuscript, short and to the point. Please see reviewer comments, and my own comments follow. I tried to mention only those that would not be redundant with the reviewers. Where a comment of mine is redundant with a reviewer comment, please indicate so and respond only to the corresponding comment made by the respective reviewer.

Text:

L28-29: Due to the placement of the version in parentheses it reads at first as if HyPhy were only the (or a) leading molecular inference platform beginning with that version, which I think is not what you mean. Consider changing this to, e.g., "To this end, beginning with version X.Y.Z we introduce ...", or something else that removes that semantic dissonance.

L33: "In addition, LEISR is additionally MPI-enabled" -> "In addition" and "additionally" are redundant.

L40: "and it then for each site, it infers" -> either "and it then for each site infers", or "and for each site it then infers".

L40: "a scaler parameter" -> Is "scaler" correct and not a typo? I'm used to "scaling parameter" or "scale parameter", though will defer to the authors whether "scaler" is the correct term here.

L43: "HyPhy will write markdown-formatted status-indicators" -> please cite Markdown, even if only by URL. (Even if this seems sad, there remain many biologists unfamiliar with Markdown format.)

L45: "written to a JSON-formatted file" -> please cite JSON, even if only by URL. See comment about Markdown :-)

L48-49: "We note that a general description of HyPhy output JSON contents is available from http://www.hyphy.org."; While I don't doubt that the information is there somewhere, I couldn't find the page that would have that information. I suggest to give a more specific URL so that users don't have to go digging around.

L64-70: I suggest to rephrase this text. There are repeated "Indeed"s, and it's a little confusing *why* users are encouraged to opt for no rate variation. It seems there are two reasons (leaking of rate variation information into branch lengths; layering LEISR as a non-parametric method over a parametric estimation method is inefficient), but these are a sentence removed and hence it's not quite clear for a reader whether that's the correct semantic parsing result.

L101-106: In theory the Conclusion section should be about the key take-aways for a reader. I suggest considering to either retitle the section, or to move some of the information here to other parts. For example, could the request to cite Rate4Site to the acknowledgments? And the sentence about earlier versions containing a more limited implementation of LEISR seems rather peripheral, perhaps that should be a footnote earlier where the 2.3.8 version is initially mentioned?"

Github repo:

- The input tree isn't in the repo, and doesn't seem to be created as an initial step in the simulation script? There's commented out code to call ape's rtree() in R, but it's not clear whether that's something that can be done or must be done first. (I think reviewer #2 observed this too.)

- The repository should be permanently archived as a snapshot prior to publication, and the archive URL be mentioned in the paper. This can be quite easily and cost-free accomplished through Zenodo (https://zenodo.org) and Figshare (https://figshare.com) through their respective Github integrations, which will subsequently link the archived snapshot to the live repository on Github. Both also issue DOIs for such deposits. See the guide here (which uses Zenodo): https://guides.github.com/activities/citable-code/. Other possibilities exist of course too.

- Consider easy to accomplish ways to state dependencies more formally. For example, for Python the requirements.txt file does allow to give source code repository URLs, including for Git, instead of PyPy packages https://pip.readthedocs.io/en/stable/reference/pip_install/#vcs-support
I realize that because there are dependencies both for the Python and for the R part, covering both ends equally well may be too challenging. That said, steps that can be taken in this direction and that are relatively easy to take should be considered.

- The Python script has several try-catch blocks that in essence make failures completely silent, as not even the error is printed. Perhaps that's justified and failure for any of these blocks is entirely inconsequential, but even if so, that's not obvious from the code, and should hence be stated in comments.

·

Basic reporting

This manuscript introduces a useful extension to HyPhy to infer relative evolutionary rates of protein and nucleotide data. This work is clearly written, provides sufficient background material, contains well-organized figures, and is nicely self-contained. I have only minor discretionary comments which I have included in the “General comments for the author” section. I feel this work is ready for publication, though I would appreciate if the authors would consider my suggestions before submitting a final version.

Experimental design

The experiments designed to test their implementation are well described. I appreciate that the authors made their code available on github, and I feel this enhances the transparency of the work.

Validity of the findings

These findings appear to be robust, and the conclusions are well stated.

Additional comments

Line 28: Is HyPhy really “the leading molecular evolution inference platform”? It may be, but this seems like a pretty bold statement considering that you don’t give a reference. Consider changing the sentence to “a leading molecular evolution inference platform”.

Line 31: This sentence makes it seem like LEISR can only be used on non-coding data.

Line 33: Change to “In addition, LEISR is MPI-enabled”

Line 50: I am not suggesting you implement this in the current version, but in future versions, it would be nice if the normalized estimates were reported as well.

Line 59: The first sentence of this paragraph is very long and hard to unpack. Consider breaking it into two sentences

Line 74: A 100 sites seems pretty small. I get that it was probably chosen to keep the run-time of simulations and rate inferences short and I have no qualms with using shorter sequences. However, adding a sentence that points out that the validity of this analysis is not dependent on protein size may help to strengthen the manuscript.

Line 81: The word “during” is repeated

Line 89: It's confusing that this paragraph starts out talking about Figure 1, then switches to Figure 2, and then goes back to talking about Figure 1. Consider moving the discussion that starts on line 94 to after the sentence that ends on line 90, so that Figure 1 is discussed then Figure 2 is discussed.

Line 90: I assume the data shown in Figure 2 is the raw data and not normalized, but maybe it should be stated.

You claim that LEISR can use nucleotide data beyond the capacity of Rate4Site, but all of the analysis appears to be done on proteins. Is Rate4Site’s ability to deal with nucleotide data so poor that a comparison between Rate4Site and LEISR cannot be done? Would inference using nucleotide data be expected to work as well as analysis of protein data? Consider sneaking a sentence in somewhere that addresses if any differences are expected when analyzing nucleotide data.

·

Basic reporting

The paper is clearly written and the background is adequate. The authors provide clear links to the actual analysis code on GitHub used for this paper.

At the beginning of paragraph 2, the authors refer to `HyPhy` as "the leading molecular evolution inference platform." There are also other widely used packages, so it might be better to change "the leading" to "a leading."

Experimental design

The design is clear and well described with a few exceptions:

1. In Approach, the authors never clearly state if LEISR estimates the tree topology or not. My impression is that it does not. If this is so, it should be made clear that the tree is provided along with the alignment.

2. For the test simulations, the authors never explain the topology / branch lengths of the tree. This seems likely to strongly affect the results in the figures (presumably the results are much better with trees with long independent branches). So some information should be given about the tree used.

3. In the introduction they note that `Rate4Site` does not handle *large alignment* or *short sequences* particularly well. It is implied that `LEISR` address both of these issues but no evidence is presented to backup this claim. This could be addressed by simply giving a table of the respective runtimes of the programs, and providing an examples where `LEISR` converges well and `Rate4Site` does not.

4. Although not obligatory, it might be nice if Figure 2 also compared the inferred rates to the *true* values used for the simulation. In the sense that this paper just describes a re-implementation of Rate4Site, this is not required. However, it could be useful to have some sense of how accurate the methods are as well.

5. It would be nice to compare `Rate4Site` and `LEISR` on an empirical data to show that they give similar results on real as well as simulated data.

Validity of the findings

The findings are clearly supported except for the missing data about the superior performance of `LEISR` on large datasets mentioned above.

Additional comments

1. The paper takes a bit of time to explain why you should normalize by the median, not the mean, but in the results they normalize by the mean. Why not normalize by the median?

---

## Round 0.2 · accepted · Accept

Kudos for an impressively quick turnaround, and I appreciate the attention to each comment. For the record, I have reviewed and find satisfactory the responses to all comments. Due to this and the minor nature of all comments, I am not sending the manuscript back out to reviewers.